# The interaction between sleep patterns and oxidative balance scores on the risk of cognitive function decline: Results from the national health and nutrition examination survey 2011–2014

**Taiwei Lou** [ID]☉, **Zhiru Zhao**☉, **Hongjin Du, Jiwei Zhang, Tian Ni**‡*, **Miaoran Wang**‡*, **Qiuyan Li**‡*

Xiyuan Hospital, China Academy of Chinese Medical Sciences, Beijing, China

☉ These authors contributed equally to this work.
‡ These authors also contributed equally to this work
* liqiuyan0928@163.com (QL); wangmiaoran123@163.com (MW); 724912801@qq.com (TN)

**Data Availability Statement:** All original data files are available from the NHANES database (https://www.cdc.gov/nchs/nhanes/index.htm). In this

## Abstract

### Background

Unhealthy sleep and exposures to oxidative factors are both associated with poor cognitive performance (PCP), but limited evidence has been found regarding the relationship between sleep patterns and oxidative factor exposures independently or jointly with the risk of PCP.

### Methods

We analyzed data from 2249 adults aged ≥60 years in the National Health and Nutrition Examination Survey (*NHANES*) database (2011–2014). Self-reported questionnaires were used to collect data on sleep duration and sleep disorder, categorizing sleep duration into three groups based on responses: short (6 hours or less per night), normal (7–8 hours per night), or long (9 hours or more per night). Sleep disorder were categorized into two groups: sleep disorder, non-sleep disorder. Oxidative balance score (*OBS*) was calculated based on 20 oxidative stress exposures related to diet and lifestyle factors, with higher scores indicating greater antioxidant exposures. Survey-based multivariable adjusted regression analyses were conducted to examine the associations between sleep patterns or *OBS* alone and in combination with overall and PCP risks.

### Results

Compared to the normal sleep duration group, the long sleep duration group had a higher risk of PCP (DSST<34) (OR = 1.91, 95% CI = 1.05–3.48, P = 0.021); while *OBS* was negatively correlated with the risk of PCP (DSST<34) [Q4 vs Q1 (OR = 0.50, 95% CI = 0.29–0.92, P = 0.004)]. There was an interaction effect between sleep patterns (sleep duration,

study, data are extracted from the following sections: Demographics Data, Dietary Data, Examination Data, Laboratory Data, and Questionnaire Data. All relevant data are within the paper and its Supporting Information files.

**Funding:** The author(s) received no specific funding for this work.

**Competing interests:** All authors declare no competing interests.

sleep disturbances) and *OBS* on PCP (DSST<34) (P = 0.002). Further stratified analysis showed that in individuals with normal sleep duration, long sleep duration, or no sleep disturbances, antioxidant exposures, compared to pro-oxidant exposures, reduced the risk of low cognitive function occurrence.

## Conclusions

In older populations, unhealthy sleep patterns (especially excessive sleep duration) and low *OBS* alone or in combination increase the risk of cognitive decline. Healthy sleep and lifestyle habits rich in antioxidant factors are crucial for protecting cognitive function in older adults.

## Introduction

Mild Cognitive Impairment (MCI) is a pathological state that serves as a transition between normal aging and dementia, where cognitive functions are affected by various factors such as blood pressure, blood lipids, lifestyle, etc [1]. Currently, the incidence of MCI is gradually increasing and as early as 2022, about 5.4 million people (22.2%) aged 71 and above in the United States were diagnosed with MCI [2]. This population has a much higher probability of further cognitive decline and progressing to dementia compared to healthy individuals, which poses significant challenges to societal economy and the quality of life of the affected population [3]. Therefore, preventing or delaying the onset and progression of MCI holds important clinical significance.

Oxidative stress is one of the important mechanisms underlying the onset and progression of MCI, and is also considered as an early event in cognitive function decline [4]. The levels of oxidative stress in the body can be influenced by the intake of exogenous antioxidants and prooxidants [5], thus individual dietary patterns and lifestyle can affect oxidative stress levels and further impact cognitive function. Sleep patterns are also a lifestyle factor, and poor sleep patterns can result in the occurrence of oxidative stress in the body [6]. Both oxidative stress and sleep patterns can lead to changes in cognitive function.

Although there have been many studies focusing on the link between different factors and cognitive function, most of them have only investigated the effect of a single factor, such as sleep and alcohol consumption, on cognitive function [7,8]. As a kind of lifestyle, sleep pattern tends to interact with other lifestyles. The Oxidative Balance Score (*OBS*) integrates the effects of dietary intake and lifestyle factors on pro-oxidants and antioxidants, and it is used to quantify the overall oxidative status. In recent years, *OBS* has been shown to be associated with various diseases, including kidney disease, cancer, and cardiovascular conditions [9–11]. Additionally, *OBS* has been found to have a close relationship with the occurrence of sleep disorders [12]. Given the strong link between cognitive function and oxidative stress, we hypothesize that *OBS* is related to cognitive decline, and we are also curious about whether *OBS* and sleep disorders jointly influence cognitive function. Furthermore, by integrating sleep patterns with other lifestyle factors and evaluating their combined impact on cognitive function, more precise clinical guidance can be obtained. Therefore, based on the *NHANES* database, we calculated the Overall Behavior Score (*OBS*) for participants based on their different lifestyles and combined this with their sleep patterns. We first evaluated the impact of *OBS* on cognitive function and further assessed the interaction between *OBS*, sleep patterns, and cognitive function.

## Methods

### Study population

The National Health and Nutrition Examination Survey (*NHANES*) [13] is a cross-sectional research database aimed at evaluating the health and nutritional status of adults and children in the United States. The data in this database are obtained from a sample survey of populations in all 50 states and the District of Columbia, with a complex, multi-stage probability design used in the survey process to ensure national representativeness of the results. Participants are randomly accessed every two years for household interviews, physical examinations, and laboratory tests. Detailed information on sampling methods and data collection has been carefully documented [14,15]. *NHANES* is conducted by the National Center for Health Statistics (NCHS), which is part of the Centers for Disease Control and Prevention (CDC), and all participants provide written informed consent.

In this study, publicly available data from the 2011–2012 and 2013–2014 survey cycles were used. We selected participants aged ≥60 years who completed cognitive assessment scales, excluded those lacking dietary, lifestyle, and sleep data, further excluded individuals with missing information, resulting in a total of 2,249 participants (1,075 males and 1,174 females) included in the study.

The original data for this study were obtained from the *NHANES* public database, exempting this study from ethical review. All participants had signed informed consent forms before participating in the *NHANES* data collection.

### Sleep patterns

During the health interview, each participant was asked to answer questions related to their sleep patterns, in the home, which were posed by trained interviewers using Computer-Assisted Personal Interview (CAPI) system. Sleep duration was collected using a self-reported questionnaire (SLD010H) asking \How much sleep do you get (hours)?" and further categorized into short (6 hours or fewer per night), normal (7–8 hours per night), or long (9 hours or more per night) [16]. Sleep disorders were assessed using a self-reported questionnaire (SLQ060) asking "Ever told by doctor have sleep disorder?" and participants were classified as having sleep disorders (Yes) or not having sleep disorders (No) based on their responses.

### Oxidative balance scores

The *OBS* is composed of 16 dietary factors and 4 lifestyle factors, representing overall oxidative balance [12,17]. The dietary factors include fiber, carotenoids, riboflavin, niacin, vitamin B6, total folate, vitamin B12, vitamin C, vitamin E, calcium, magnesium, zinc, copper, selenium, total fat, and iron; while the lifestyle factors include BMI, physical activity, alcohol consumption, and smoking status. Based on their oxidative properties, the 20 factors are categorized into pro-oxidative factors (total fat, iron, alcohol intake, BMI, and smoking) and anti-oxidative factors (the remaining 15 factors).

Dietary intake data was assessed using 24-hour dietary recall interviews (24 HR) based on the University of Texas Food Intake Analysis System and the USDA National Nutrition Survey Database at mobile examination centers. Sixteen dietary factors and alcohol consumption were calculated based on two 24-hour dietary recall interviews (the first dietary recall interview was conducted in person at the Mobile Examination Center (MEC), and the second interview was conducted via phone 3 to 10 days later). Each dietary component was categorized into three groups based on gender and tertiles, and assigned scores; the highest tertile of antioxidant factors received 2 points, the middle tertile received 1 point, and the lowest tertile received 0 points; the scoring for pro-oxidative factors was the opposite. Details of the scoring system are presented in the (S1 Table).

BMI (Body Mass Index) was collected by trained health technicians at Mobile Examination Centers (MEC). It is calculated by dividing weight in kilograms by height in meters squared, rounded to one decimal place. Individuals with a normal weight ($<25$ kg/m$^2$) received 2 points, overweight individuals (25–30 kg/m$^2$) received 1 point, and obese individuals ($\geq$30 kg/m$^2$) received 0 points. Physical activity is expressed in weekly *MET*-minutes [18], calculated by summing the total time spent on walking/cycling, moderate work activities, vigorous work activities, moderate recreational activities, and vigorous recreational activities within a week, and then multiplying by the suggested *MET* scores from *NHANES*. Low intensity ($<500$ *MET* minutes per week) was assigned 0 points, moderate intensity (500–1000 *MET* minutes per week) was assigned 1 point, and high intensity ($>1000$ *MET* minutes per week) was assigned 2 points. Smoking status was determined by the following questions: "Smoked at least 100 cigarettes in life?" and " Do you now smoke cigarettes?" Current smokers (Yes, Yes) were assigned 0 points, former smokers (Yes, No) were assigned 1 point, and nonsmokers (No, No) were assigned 2 points (answers in parentheses). Questions related to physical activity and smoking status were asked before the physical examination, in the home, using the Computer-Assisted Personal Interview (CAPI) system.

The *OBS* score is calculated by summing the scores of the above factors, with the level of antioxidant exposures increasing as the score increases. *OBS* is divided into four quartile groups, with quartiles Q1 and Q2 considered as the pro-oxidation group, and Q3 and Q4 considered as the anti-oxidation group.

## Cognitive functioning

At the end of the face-to-face private interview at the Mobile Examination Center (MEC interview), cognitive function tests are evaluated by well-trained interviewers. Symbol Digit Substitution Test (DSST), CERAD Word Learning Subtest (CERAD W-L), and Animal Fluency Test (AF) are included in the *NHANES* cognitive assessment module section and their results are also included in this study [19].

The Digit Symbol Substitution Test (DSST) is a performance subtest in the Wechsler Adult Intelligence Scale (WAIS III), which relies on processing speed, sustained attention, and working memory. It includes 9 pairs of numbers and symbols. Participants have 2 minutes to copy the corresponding symbols in the adjacent 133 boxes next to the numbers, and their score is the total number of correct matches. The CERAD Word List Learning (CERAD W-L) subtest assesses immediate and delayed learning ability for new verbal information (memory domain). The test includes three consecutive learning trials and a delayed recall. In the learning trials, participants are instructed to loudly read out 10 unrelated words, one at a time, as they are presented. After the introduction of the words, participants immediately recall as many words as possible. Across the three learning trials, the order of the 10 words is changed in each trial. Delayed word recall occurs approximately 8–10 minutes after the initiation of the word-learning trials. The Animal Fluency Test evaluates category fluency of language, which is a component of executive function. Participants are asked to name as many animals as possible within one minute. Each named animal earns one point.

Based on previous research, PCP (Poor Cognitive Performance) is defined as DSST $<34$ [20]; the following critical values for additional cognitive tests were used to test the sensitivity of PCP in this study [21]: $<17$ for CERAD-WL, $<5$ for CERAD-DR, and $<14$ for AF.

## Covariates

Using a standardized questionnaire to collect research variable results such as gender, age, race/ethnicity, education level, marital status, economic income, and health conditions from

family interviews. Race is categorized as *Mexican American*, *other Hispanic*, *Non-Hispanic White*, *Non-Hispanic Black*, *other race (Including Multi-Racial)*. Education level is divided into *Less than 9th grade*, *9-11th grade (Includes 12th grade with no diploma)*, *High school graduate/ GED or equivalent*, *Some college/AA degree*, *College graduate or above*. Marital status is categorized as *Married/Living with partner*, *Widowed/Divorced/Separated*, *Never married*. According to PIR (Ratio of family income to poverty), economic income is divided into *PIR < 1.3*, *1.3 ≤ PIR < 3.5, PIR ≥ 3.5*. Diabetes is defined as self-reported doctor's diagnosis of DM (Diabetes Mellitus), glycated hemoglobin A1c (HbA1c) ≥ 6.5%, use of insulin or antidiabetic medications, fasting blood glucose ≥ 7.0 mmol/L, random blood glucose ≥ 11.1 mmol/L, or oral glucose tolerance test (OGTT) ≥ 11.1 mmol/L. Hyperlipidemia is defined as total cholesterol ≥ 200 mg/dL, triglycerides ≥ 150 mg/dL, LDL cholesterol ≥ 130 mg/dL, and HDL cholesterol < 40 mg/dL, or participants reporting the use of lipid-lowering medications are also classified as having dyslipidemia. Depression is defined as a total score ≥ 10 on the Patient Health Questionnaire 9 (PHQ-9) in self-administered questionnaires. Hypertension is defined as an average of three blood pressure measurements with systolic blood pressure > 140 mmHg, diastolic blood pressure > 90 mmHg, or self-reported doctor diagnosis of hypertension; participants reporting the use of antihypertensive prescriptions and medications are also classified as having hypertension.

## Statistical analysis

Due to the complex sampling design of *NHANES*, sample weights, clustering, and stratification were applied to all analyses in this study. The original two-year sample weights were divided by 2 in this study, and the detailed steps for handling the remaining *NHANES* weights have been documented in previous studies [14,16].

Continuous variables were presented as median and interquartile range, while categorical variables were presented as frequencies and percentages. Differences among different cognitive groups, *OBS* quartile groups, or sleep patterns were described using Wilcoxon rank-sum test and chi-squared tests. Weighted multivariable logistic regression analysis was conducted to study the relationship between sleep patterns, *OBS*, and the risk of cognitive impairment, followed by further stratified analysis on the impact of the interaction between sleep patterns and *OBS* on the risk of cognitive decline.

Three models were used in this study to adjust for covariates as confounding effects. Model 1 adjusted for age, sex, and race. Model 2 further adjusted for education level, marital status, and family income based on Model 1. Model 3 further adjusted for diabetes, hyperlipidemia, depression, and hypertension based on Model 2. Additionally, stratified analysis was conducted to explore the role of covariates in the interaction between sleep patterns and *OBS* on cognitive performance.

## Results

### Population characteristics

This study involved a total of 2,249 participants, including 1,075 males and 1,174 females, with an average age of 68 years. Among the participants, 494 (12%) exhibited cognitive impairment. Cognitive impairment was more likely to occur in older individuals, other Hispanic ethnicities, those with lower education levels, unmarried individuals, those with lower incomes, individuals with diabetes, depression, and hypertension. Importantly, individuals with cognitive impairment tended to have lower *OBS* and abnormal nighttime sleep duration. Furthermore, compared to short sleep duration, longer sleep duration appeared to have a greater impact on cognition (See Table 1) (S4 Table).

**Table 1. Characteristics of weighted study participants according to Cognitive Performance, NHANES 2011 to 2014 (Non-weighted n = 2249).**

| Characteristic | N[1] | Cognitive Performance (DSST score) | | | P Value[3] |
|---|---|---|---|---|---|
| | | Overall, N = 2249 (100%)[2] | Normal(≥34), N = 1755 (88%)[2] | Poor(<34), N = 494 (12%)[2] | |
| **Gender** | 43,432,366 | | | | 0.7 |
| *Male* | | 1,075 (45%) | 805 (45%) | 270 (46%) | |
| *Female* | | 1,174 (55%) | 950 (55%) | 224 (54%) | |
| **Age (years)** | 43,432,366 | | | | **<0.001***\* |
| *60–65* | | 833.0 (40.1%) | 716.0 (43.3%) | 117.0 (17.2%) | |
| *66–70* | | 509.0 (22.4%) | 401.0 (23.0%) | 108.0 (17.8%) | |
| *71–75* | | 366.0 (15.9%) | 279.0 (15.6%) | 87.0 (17.6%) | |
| *76–80* | | 541.0 (21.7%) | 359.0 (18.1%) | 182.0 (47.4%) | |
| **Race** | 43,432,366 | | | | **<0.001***\* |
| *Mexican American* | | 186.00 (3.02%) | 122.00 (2.25%) | 64.00 (8.56%) | |
| *Other Hispanic* | | 204.00 (3.10%) | 120.00 (2.05%) | 84.00 (10.63%) | |
| *Non-Hispanic White* | | 1,153.00 (81.83%) | 997.00 (85.19%) | 156.00 (57.59%) | |
| *Non-Hispanic Black* | | 525.00 (7.81%) | 351.00 (5.97%) | 174.00 (21.09%) | |
| *Other Race(Including Multi-Racial)* | | 181.00 (4.25%) | 165.00 (4.54%) | 16.00 (2.12%) | |
| **Education level** | 43,432,366 | | | | **<0.001***\* |
| *Less than 9th grade* | | 209 (4.9%) | 53 (2.1%) | 156 (25%) | |
| *9-11th grade (Includes 12th grade with no diploma)* | | 299 (9.8%) | 191 (8.2%) | 108 (21%) | |
| *High school graduate/GED or equivalent* | | 526 (21%) | 408 (20%) | 118 (27%) | |
| *Some college/AA degree* | | 660 (32%) | 582 (34%) | 78 (19%) | |
| *College graduate or above* | | 553 (32%) | 521 (35%) | 32 (8.2%) | |
| *Don't know/Refused* | | 2 (<0.1%) | 0 (0%) | 2 (0.2%) | |
| **Marital status** | 43,432,366 | | | | **<0.001***\* |
| *Married/Living with partner* | | 1,315 (66%) | 1,075 (68%) | 240 (51%) | |
| *Widowed/Divorced/Separated* | | 808 (30%) | 581 (28%) | 227 (44%) | |
| *Never married* | | 125 (4.2%) | 98 (4.0%) | 27 (5.1%) | |
| *Don't know/Refused* | | 1 (<0.1%) | 1 (<0.1%) | 0 (0%) | |
| **Ratio of family income to poverty** | 43,432,366 | | | | **<0.001***\* |
| *PIR<1.3* | | 622 (16%) | 381 (13%) | 241 (39%) | |
| *1.3≤PIR<3.5* | | 884 (39%) | 694 (38%) | 190 (46%) | |
| *PIR≥3.5* | | 743 (45%) | 680 (49%) | 63 (15%) | |
| **Diabetes** | 43,432,366 | | | | **<0.001***\* |
| *Yes* | | 736 (26%) | 514 (24%) | 222 (43%) | |
| *No* | | 1,513 (74%) | 1,241 (76%) | 272 (57%) | |
| **Hyperlipidemia** | 43,432,366 | | | | 0.5 |
| *Yes* | | 1,876 (84%) | 1,463 (84%) | 413 (85%) | |
| *No* | | 373 (16%) | 292 (16%) | 81 (15%) | |
| **Depression** | 43,432,366 | | | | **<0.001***\* |
| *Yes* | | 201 (7.2%) | 129 (6.2%) | 72 (15%) | |
| *No* | | 2,048 (93%) | 1,626 (94%) | 422 (85%) | |
| **Hypertension** | 43,432,366 | | | | **<0.001***\* |
| *Yes* | | 1,576 (65%) | 1,178 (63%) | 398 (82%) | |
| *No* | | 673 (35%) | 577 (37%) | 96 (18%) | |
| **Sleep duration** | 43,432,366 | | | | **<0.001***\* |
| *Normal(7-8h)* | | 1,269 (62%) | 1,002 (63%) | 267 (53%) | |
| *Short Sleep(≤6h)* | | 748 (27%) | 588 (27%) | 160 (29%) | |

*(Continued)*

**Table 1.** (Continued)

| Characteristic | N[1] | Cognitive Performance (DSST score) | | | P Value[3] |
|---|---|---|---|---|---|
| | | Overall, N = 2249 (100%)[2] | Normal(≥34), N = 1755 (88%)[2] | Poor(<34), N = 494 (12%)[2] | |
| *Long Sleep(≥9h)* | | 232 (11%) | 165 (9.6%) | 67 (18%) | |
| **Sleep disorder** | 43,432,366 | | | | 0.5 |
| *Yes* | | 276 (12%) | 226 (12%) | 50 (11%) | |
| *No* | | 1,973 (88%) | 1,529 (88%) | 444 (89%) | |
| **OBS** | 43,432,366 | | | | **<0.001***\* |
| *Q1* | | 593 (22%) | 400 (20%) | 193 (38%) | |
| *Q2* | | 604 (25%) | 466 (25%) | 138 (28%) | |
| *Q3* | | 544 (26%) | 443 (27%) | 101 (19%) | |
| *Q4* | | 508 (27%) | 446 (28%) | 62 (15%) | |

[1]N not Missing.

[2]median (IQR) for continuous; n (%) for categorical.

[3]Wilcoxon rank-sum test for complex survey samples; chi-squared test with Rao & Scott's second-order correction.

PIR–poverty-income ratio, OBS–oxidative balance score, Q–quartile.

*$P < 0.05$

**$P < 0.01$

***$P < 0.001$.

Based on sleep patterns, individuals with sleep disorders tend to be male, younger in age, with diabetes, high cholesterol, and depression. Those with longer sleep duration are more likely to be male, older in age, non-Hispanic white, lower educated, unmarried, and hypertensive (S2 Table). In terms of *OBS* scores, higher *OBS* scores are more common among females, non-Hispanic white individuals, those in relationships, with high income, and without other illnesses (S3 Table).

## Association of sleep patterns with risk of cognitive decline

After all covariates were adjusted (Model 3), compared with normal-sleeper populations, long-sleepers had an elevated risk of cognitive decline—DSST<34 (OR = 1.91, 95% CI = 1.05–3.48, P = 0.021), CERAD-DR<5 (OR = 1.83, 95% CI = 1.11–3.04, P<0.001), and AF<14 (OR = 1.89, 95% CI = 1.13–3.16, P = 0.014); and the risk of CERAD-WL<17, although the risk of CERAD-WL<17 was significant in the original model (OR = 1.64, 95%CI = 1.11–2.43, P = 0.035), the significance disappeared as covariates were adjusted (See Table 2) (S5 Table).

No significant differences in the risk of cognitive decline between the short sleep duration group and the normal sleep duration group were observed. Similarly, there was no significant difference in the risk of cognitive decline between individuals with sleep disorders and those without sleep disorders (S5 Table).

## Association of OBS with risk of cognitive decline

Using DSST scores as an example, an inverse correlation was observed between higher *OBS* scores and the risk of cognitive impairment—Q3 vs Q1 (OR = 0.50, 95%CI = 0.27–0.91, P = 0.004), and Q4 vs Q1 (OR = 0.50, 95%CI = 0.29–0.92, P = 0.004). This negative correlation was also observed in CERAD-WL, where CERAD-WL: Q4 vs Q1 (OR = 0.50, 95%CI = 0.29–0.92, P<0.001). In the AF group, there was a significant difference in cognition between Q3 vs Q1 (OR = 0.56, 95%CI = 0.36–0.85, P<0.001), but this difference disappeared in the Q4 group.

**Table 2. Weighted odds ratios with 95% CI for the associations between sleep patterns and PCP (DSST<34).**

| Characteristic | Crude model | | Model 1 | | Model 2 | | Model 3 | |
|---|---|---|---|---|---|---|---|---|
| | OR[1](95% CI[1]) | P-value | OR[1](95% CI[1]) | P-value | OR[1](95% CI[1]) | P-value | OR[1](95% CI[1]) | P-value |
| **Sleep duration** | | **0.002**** | | **0.006**** | | **0.023*** | | **0.021*** |
| *Normal(7-8h)* | — | | — | | — | | — | |
| *Short Sleep(≤6h)* | 1.29(0.99, 1.68) | | 1.11(0.76, 1.62) | | 0.99(0.64, 1.51) | | 0.89(0.56, 1.42) | |
| *Long Sleep(≥9h)* | 2.18(1.38, 3.44) | | 2.02(1.27, 3.22) | | 1.95(1.12, 3.41) | | 1.91(1.05, 3.48) | |
| **Sleep disorder** | | 0.5 | | 0.7 | | 0.6 | | 0.8 |
| *Yes* | — | | — | | — | | — | |
| *No* | 1.14(0.77, 1.67) | | 0.92(0.61, 1.37) | | 0.89(0.54, 1.46) | | 1.08(0.62, 1.89) | |

[1]OR = Odds Ratio, CI = Confidence Interval.

*P < 0.05

**P<0.01

***P<0.001.

However, there was no statistically significant difference in CERAD-DR scores between *OBS* groups (See Table 3) (S6 Table).

## The association between sleep patterns and the risk of cognitive impairment in conjunction with OBS

Participants were divided into different subgroups based on their sleep patterns and *OBS*. Compared to the group with normal sleep duration and antioxidant factors, the risk of cognitive impairment increased in other groups, especially in the group with long sleep duration and pro-oxidative factors (DSST, CERAD-WL, AF). Compared to the group with normal sleep (without sleep disorders) and antioxidant factors, the risk of cognitive impairment increased in the group with sleep disorders and pro-oxidative factors (DSST, CERAD-WL, AF) (Fig 1) (S7 Table).

**Table 3. Weighted odds ratios with 95% CI for the associations between OBS and PCP (DSST<34).**

| Characteristic | Crude model | | Model 1 | | Model 2 | | Model 3 | |
|---|---|---|---|---|---|---|---|---|
| | OR[1](95% CI[1]) | P-value | OR[1](95% CI[1]) | P-value | OR[1](95% CI[1]) | P-value | OR[1](95% CI[1]) | P-value |
| **OBS (quartile)** | | **<0.001*** | | **<0.001*** | | **0.001**** | | **0.004**** |
| *Q1* | — | | — | | — | | — | |
| *Q2* | 0.58(0.42, 0.80) | | 0.58(0.39, 0.87) | | 0.66(0.41, 1.07) | | 0.70(0.42, 1.18) | |
| *Q3* | 0.35(0.22, 0.55) | | 0.36(0.22, 0.60) | | 0.46(0.27, 0.80) | | 0.50(0.27, 0.91) | |
| *Q4* | 0.26(0.16, 0.43) | | 0.30(0.19, 0.48) | | 0.48(0.27, 0.83) | | 0.50(0.29, 0.92) | |
| **OBS (median)** | | **<0.001*** | | **<0.001*** | | **<0.001*** | | **<0.001*** |
| *Pro-oxidative* | — | | — | | — | | — | |
| *Anti-oxidative* | 0.40(0.30, 0.54) | | 0.44(0.34, 0.57) | | 0.58(0.43, 0.79) | | 0.60(0.44, 0.83) | |

[1]OR = Odds Ratio, CI = Confidence Interval.

*P < 0.05

**P<0.01

***P<0.001.

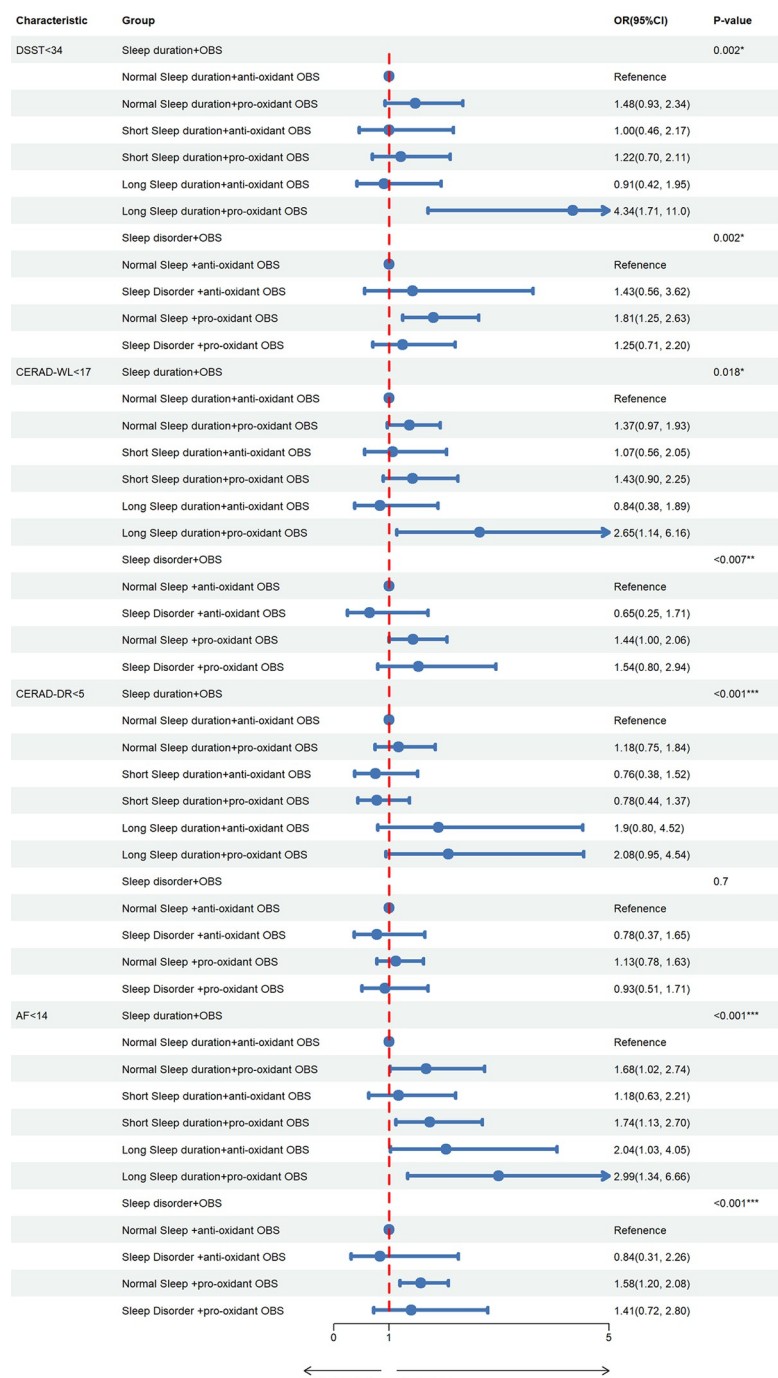

**Fig 1. Multiple logistic regression on the association between comprehensive sleep patterns and OBS and PCP.**
Adjusted for model 3. OR = Odds Ratio, CI = Confidence Interval. *$P < 0.05$, **$P < 0.01$, ***$P < 0.001$.

Sleep patterns were used as stratification criteria to further investigate the combined effects of sleep patterns and *OBS* (antioxidant, pro-oxidant factors) on the risk of cognitive impairment. Taking DSST scores as an example, in populations with normal sleep duration, long sleep duration, or no sleep disorders, exposures to antioxidant factors was associated with a decreased risk of low cognitive function compared to exposures to pro-oxidant factors.

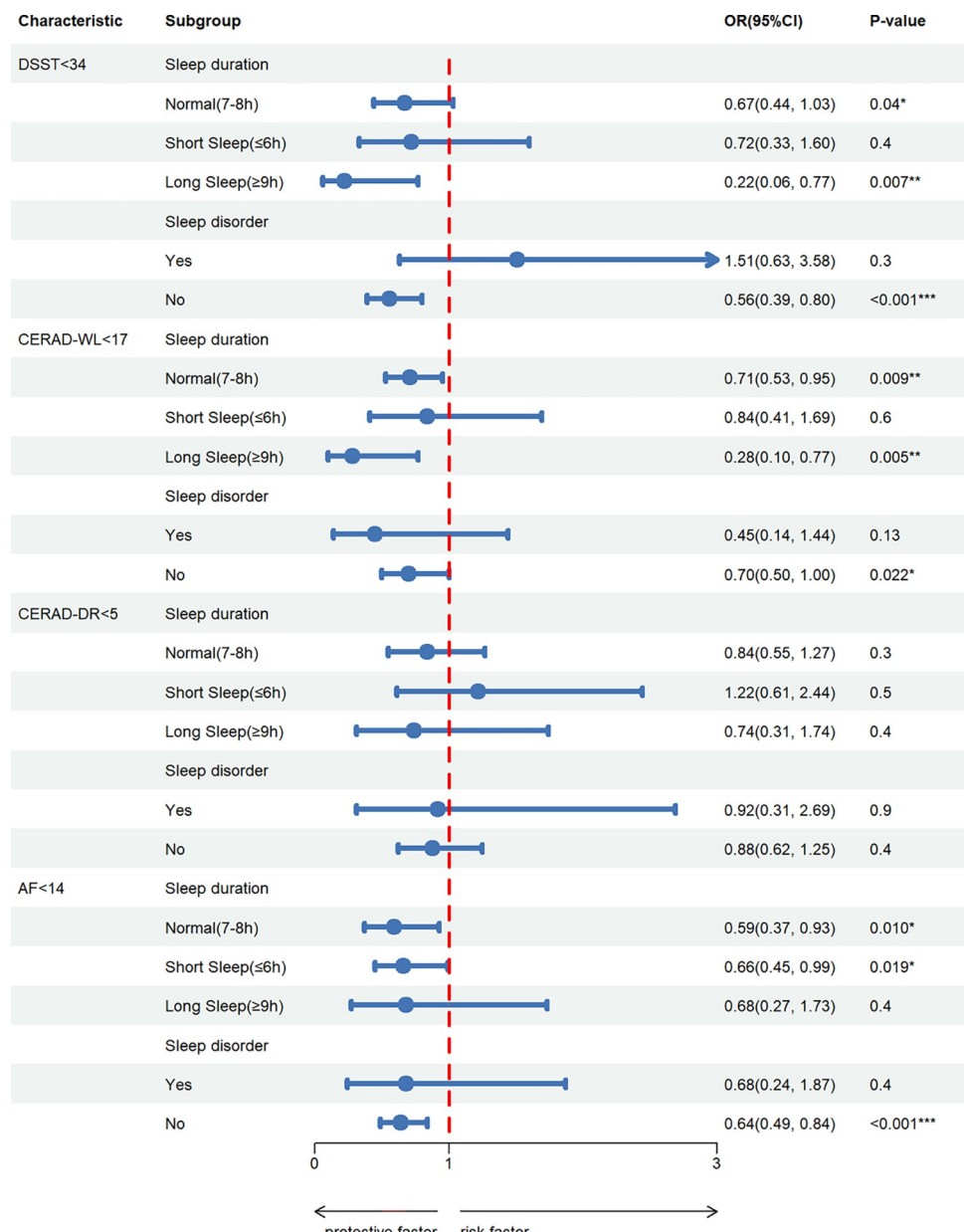

**Fig 2. Forest plot of stratified multiple logistic regression analysis on the association between anti-oxidant OBS and PCP compared to pro-oxidant OBS.** Adjusted for model 3. OR = Odds Ratio, CI = Confidence Interval. *$P < 0.05$, **$P < 0.01$, ***$P < 0.001$.

Similar results were found in CERAD-WL scores. In populations with short sleep duration exposed to antioxidant factors, the risk of decreased AF scores was reduced. No significant correlation was found in CERAD-DR scores (Fig 2) (S8 Table).

## Discussion

In this study, we found that sleep patterns and *OBS* can independently or jointly affect the risk of cognitive impairment. Participants with long sleep duration and exposures to pro-oxidant factors had an increased risk of cognitive impairment (DSST, CERAD-WL, AF). Further

stratified analysis showed that in populations with normal sleep duration, long sleep duration, or no sleep disorders, exposures to antioxidant factors was associated with a decreased risk of PCP compared to exposures to pro-oxidant factors.

Changes in sleep and PCP are commonly found in the elderly population. Many previous studies have indicated that alterations in sleep structure and quality, such as REM sleep, sleep efficiency, sleep latency, and sleep duration, seem to be risk factors for PCP [22–24]. Multiple research results suggest a positive correlation between the occurrence of sleep disorders, prolonged sleep duration, and PCP [25–28]. This study further supports this conclusion. Some scholars have suggested that the process of PCP caused by sleep disorders may be related to mechanisms such as elevated levels of amyloid-beta, abnormal tau protein phosphorylation, neuroinflammation, neurodegeneration, and oxidative stress [23,29]. Oxidative stress is also considered a significant factor in the occurrence of these mechanisms. Sleep disorders promote the accumulation of reactive oxygen species, impair neural function and development, leading to PCP [6,30–32]. In sleep-deprived mice, activated microglia and astrocytes in the brain were found, accompanied by a decrease in antioxidant enzyme levels [33]. A cross-sectional study involving 6300 participants found a negative correlation between *OBS* and sleep disorders, and a positive correlation with sleep duration [12]. Another cross-sectional study found that higher *OBS* in short sleepers was associated with lower all-cause mortality [34].

Oxidative stress plays an important role in the process of cognitive decline [35–37], and it is also one of the targets for various drugs to reduce neuronal damage and improve cognitive function [38–40]. However, the relationship between *OBS* as an important score for individual oxidative factor exposures and cognitive decline has not yet been determined. Therefore, for the first time in this study, we explored the relationship between *OBS* and cognitive function, and further investigated the combined impact of sleep patterns and *OBS* on the risk of cognitive decline.

Our results indicate that participants with long sleep duration and exposures to oxidative factors are more likely to experience cognitive decline. Additionally, in populations with normal sleep duration, long sleep duration, or no sleep disorders, oxidative factors seem to have a more pronounced impact on cognitive function (DSST, CERAD-WL). However, the impact of oxidative factors on language fluency (AF) in participants with normal sleep duration, short sleep duration, or no sleep disorders is greater, reflecting the participants' executive function. Regarding this phenomenon, several hypotheses have been proposed. One hypothesis suggests that the differences are not causally related, and the differences stem from potential confounding factors that have not been considered, such as cognitive-related brain disorders. Another hypothesis proposes that unhealthy sleep patterns disrupt the regularity of life, leading to changes in oxidative stress, as alterations in sleep patterns may result in changes in diet structure, exercise routines, causing variations in oxidative stress levels. Alternatively, low oxidative stress may disrupt normal sleep rhythms, and our study also found a connection between low oxidative stress and sleep disorders. Furthermore, participants with or without sleep disorders showed no differences in cognitive function. However, participants without sleep disorders were more sensitive to oxidative factor exposures in terms of cognitive function, suggesting that sleep disorders themselves may act as an oxidative factor leading to cognitive decline. The study also found that short sleep duration was not significantly associated with cognitive decline, and the impact of insomnia and oxidative factors on participants' delayed memory (CERAD-DR) appeared to be relatively small. This may be due to the fact that most participants were elderly individuals with reduced physiological sleep time accompanied by memory decline or other confounding factors. However, it is worth noting that our results emphasize the important impact of healthy sleep patterns and lifestyle on cognitive function protection in

elderly individuals. Maintaining healthy sleep and a healthy lifestyle can help prevent cognitive decline.

This study also has the following limitations. Firstly, although some confounding variables have been adjusted through multiple logistic regression, there may still be unaccounted confounding factors that affect the accuracy of the results. Secondly, this study is a cross-sectional survey, which cannot determine the exact temporal causal relationship between sleep, *OBS*, and cognitive function. Moreover, the diagnosis of insomnia in this study relies on self-reported questionnaires, which are susceptible to subjective factors of the participants, and the level of cognitive function of the participants may also affect the accuracy of questionnaire responses. In future studies, integrating participants' questionnaire responses with polysomnography tests may yield more accurate conclusions. In this study, using the R language "mediation" package, we investigated whether there is a mediating effect between sleep, *OBS*, and cognitive function, but the analysis was limited by complex weighting. In the unweighted analysis, no mediating effect was found among the three factors. Finally, we only used data provided by participants from certain regions of the United States, further research is needed to explore the situation of participants from other regions or countries.

## Conclusions

In older populations, unhealthy sleep patterns (especially excessive sleep duration) and low *OBS* alone or in combination increase the risk of cognitive decline. Healthy sleep and lifestyle habits rich in antioxidant factors are crucial for protecting cognitive function in older adults.

## Supporting information

**S1 Table. Components of the Oxidative Balance Score (OBS).**
(DOCX)

**S2 Table. Characteristics of weighted study participants according to sleep patterns, NHANES 2011 to 2014 (Non-weighted n = 2249).**
(DOCX)

**S3 Table. Characteristics of weighted study participants according to OBS, NHANES 2011 to 2014 (Non-weighted n = 2249).**
(DOCX)

**S4 Table. Characteristics of weighted study participants according to Cognitive Performance (CERAD, AF), NHANES 2011 to 2014 (Non-weighted n = 2249).**
(DOCX)

**S5 Table. Weighted odds ratios with 95% CI for the associations between sleep patterns and cognitive performance (PCP).**
(DOCX)

**S6 Table. Weighted odds ratios with 95% CI for the associations between OBS and cognitive performance (PCP).**
(DOCX)

**S7 Table. Weighted odds ratios with 95% CI from logistic regression analyses of the association between combined sleep patterns and OBS with cognitive performance (PCP).**
(DOCX)

**S8 Table. Weighted odds ratios with 95% CI from stratified analyses of the associations of anti-oxidative OBS with cognitive performance (PCP) compared with pro-oxidant OBS.**
(DOCX)

## Author Contributions

**Conceptualization:** Tian Ni, Qiuyan Li.

**Data curation:** Taiwei Lou, Hongjin Du, Jiwei Zhang, Miaoran Wang.

**Methodology:** Miaoran Wang.

**Writing – original draft:** Taiwei Lou, Zhiru Zhao.

**Writing – review & editing:** Tian Ni, Qiuyan Li.

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
