## [Decision Letter · Decision Letter 0]

8 Oct 2024

PONE-D-24-12132The interaction between sleep patterns and oxidative balance scores on the risk of cognitive function decline: Results from the National Health and Nutrition Examination Survey 2011-2014.PLOS ONE

Dear Dr. Lou,

Thank you for submitting your manuscript to PLOS ONE. After careful consideration, we feel that it has merit but does not fully meet PLOS ONE’s publication criteria as it currently stands. Therefore, we invite you to submit a revised version of the manuscript that addresses the points raised during the review process.

We look forward to receiving your revised manuscript.

Kind regards,

Faizul Hasan, PhD

Academic Editor

PLOS ONE

Journal Requirements:

“All authors declare no competing interests.”

4. We notice that your supplementary [Table S1 and Tables 1-8] are included in the manuscript file. Please remove them and upload them with the file type 'Supporting Information'. Please ensure that each Supporting Information file has a legend listed in the manuscript after the references list.

Additional Editor Comments:

Thank you for your submission. After reviewing your manuscript, I believe that it requires minor revisions. Please ensure that you carefully revise and recheck your work according to the journal's guidelines. Addressing these points will enhance the clarity and quality of your submission.

If you have any questions or need further clarification on specific aspects, feel free to reach out.

Reviewers' comments:

Reviewer's Responses to Questions

**Comments to the Author**

1. Is the manuscript technically sound, and do the data support the conclusions?

Reviewer #1: Yes

Reviewer #2: Yes

2. Has the statistical analysis been performed appropriately and rigorously? 

Reviewer #1: Yes

Reviewer #2: Yes

3. Have the authors made all data underlying the findings in their manuscript fully available?

Reviewer #1: Yes

Reviewer #2: Yes

4. Is the manuscript presented in an intelligible fashion and written in standard English?

Reviewer #1: Yes

Reviewer #2: Yes

5. Review Comments to the Author

Reviewer #1: It is a very nice study with a complex but well-designed statistical analysis. Just adopt the same presentation for all tables of results. Tables 1 and 2 are not presented in the same way as the others tables: the titles of the columns are indeed not repeated at the top of each new page.

Reviewer #2: Comments for the authors:

Thank you for presenting the evidence on the interaction between sleep patterns and oxidative balance scores in relation to cognitive function risk. While I find your results promising, I have provided some feedback to help clarify certain aspects of the paper for readers. My comments are outlined below:

1. Although the introduction provides a brief overview of the study's aim to assess the interaction between OBS, sleep patterns, and cognitive function, it lacks a clearly defined research hypothesis or specific objectives. To strengthen the introduction, it should conclude with a precise hypothesis that clearly outlines what the researchers intend to test or demonstrate in the study. Please include this information for clarity.

2. It's preferable not to use abbreviations in your keywords.

3. According the discussion part, the use of self-reported questionnaires to diagnose insomnia and assess cognitive function is mentioned as a limitation, but the effect of this bias on the study’s results is not fully addressed. Since self-reported data can be subjective, especially in elderly participants, this issue should be explored more, and alternative objective methods could be suggested for future research. Please add more information.

6. PLOS authors have the option to publish the peer review history of their article (what does this mean?). If published, this will include your full peer review and any attached files.

Reviewer #1: **Yes: **Olivier Coste

Reviewer #2: No

---

## [Author Response · Author response to Decision Letter 0]

27 Oct 2024

Dear Esteemed Reviewers and Editor,

I hope this message finds you well.

I am writing to express my heartfelt gratitude for the valuable feedback from you, the esteemed reviewers and editor, regarding my manuscript titled “The Interaction Between Sleep Patterns and Oxidative Balance Scores on the Risk of Cognitive Function Decline: Results from the National Health and Nutrition Examination Survey2011-2014.” I am truly thankful for your dedication and insightful comments.

In response to your suggestions, I have made the following revisions to the manuscript:

1. In accordance with PLOS ONE’s formatting requirements, I have revised the overall format of the manuscript, including author attribution, title formatting and size, citation format for figures and tables, and reference formatting. Additionally, I have incorporated the supporting information directly into the manuscript. We made adjustments to the table format to ensure that the headers repeat across pages for the reader's convenience. If you notice any remaining formatting inconsistencies in the revised manuscript, please do not hesitate to contact me, and I will be more than happy to correct any errors.

2. I have moved the ethics statement to the methodology section.

3. We have removed the table from the manuscript that was also repeated in the supplementary materials.

4. I have checked and updated the references to ensure that there are no retracted publications included.

5. In the introduction, I enhanced the discussion of the oxidative balance score (OBS) and clearly presented the hypotheses underpinning our research questions, thereby improving the logical flow of the paper.

6. I have removed the abbreviations from the keywords, except for “NHANES,” as we noted that many other articles have adopted a similar approach.

7. In the discussion section, I have provided additional information pertaining to the limitation regarding “self-reported data may be subjective” and offered potential methods for addressing this limitation.

Finally, these are the numerous revisions made to the manuscript. I sincerely appreciate the diligence of all reviewers and kindly request any further feedback you may have on our manuscript.

Warm regards,

Taiwei Lou

---

## [Editor Report · Decision Letter 1]

31 Oct 2024

The interaction between sleep patterns and oxidative balance scores on the risk of cognitive function decline: Results from the National Health and Nutrition Examination Survey 2011-2014.

PONE-D-24-12132R1

Dear Dr. Lou,

We’re pleased to inform you that your manuscript has been judged scientifically suitable for publication and will be formally accepted for publication once it meets all outstanding technical requirements.

Kind regards,

Faizul Hasan, PhD

Academic Editor

PLOS ONE
---

## [Editor Report · Acceptance letter]

12 Dec 2024

PONE-D-24-12132R1 

PLOS ONE

Dear Dr. Lou, 

I'm pleased to inform you that your manuscript has been deemed suitable for publication in PLOS ONE. Congratulations! Your manuscript is now being handed over to our production team.

Kind regards, 

on behalf of

Dr. Faizul Hasan 

Academic Editor

PLOS ONE